# THE UNUSUAL EFFECTIVENESS OF AVERAGING IN GAN TRAINING

**Yasin Yazıcı**[*]
Nanyang Technological University (NTU)

**Chuan-Sheng Foo**
Institute for Infocomm Research, A*STAR

**Stefan Winkler**
National University of Singapore (NUS)

**Kim-Hui Yap**
Nanyang Technological University (NTU)

**Georgios Piliouras**[†]
Singapore University of Technology and Design

**Vijay Chandrasekhar**[†]
Institute for Infocomm Research, A*STAR

## ABSTRACT

We examine two different techniques for parameter averaging in GAN training. Moving Average (MA) computes the time-average of parameters, whereas Exponential Moving Average (EMA) computes an exponentially discounted sum. Whilst MA is known to lead to convergence in bilinear settings, we provide the – to our knowledge – first theoretical arguments in support of EMA. We show that EMA converges to limit cycles around the equilibrium with vanishing amplitude as the discount parameter approaches one for simple bilinear games and also enhances the stability of general GAN training. We establish experimentally that both techniques are strikingly effective in the non-convex-concave GAN setting as well. Both improve inception and FID scores on different architectures and for different GAN objectives. We provide comprehensive experimental results across a range of datasets – mixture of Gaussians, CIFAR-10, STL-10, CelebA and ImageNet – to demonstrate its effectiveness. We achieve state-of-the-art results on CIFAR-10 and produce clean CelebA face images.[1]

## 1 INTRODUCTION

Generative Adversarial Networks (GANs) (Goodfellow et al., 2014) are two-player zero-sum games. They are known to be notoriously hard to train, unstable, and often don't converge. There has been a lot of recent work to improve the stability of GANs by addressing various underlying issues. Searching for a more stable function family (Radford et al., 2015) and utilizing better objectives (Arjovsky et al., 2017; Gulrajani et al., 2017) have been explored for training and instability issues. For tackling non-convergence, regularizing objective functions (Mescheder et al., 2017; Salimans et al., 2016), two time-scale techniques (Heusel et al., 2017) and extra-gradient (optimistic) methods (Daskalakis et al., 2018; Mertikopoulos et al., 2019) have been studied.

One reason for non-convergence is cycling around an optimal solution (Mertikopoulos et al., 2018; Papadimitriou & Piliouras, 2018), or even slow outward spiraling (Bailey & Piliouras, 2018). There might be multiple reasons for this cycling behavior: (i) eigenvalues of the Jacobian of the gradient vector field have zero real part and a large imaginary part (Mescheder et al., 2017). (ii) Gradients of the discriminator may not move the optimization in the right direction and may need to be updated many times to do so (Arjovsky et al., 2017). (iii) Sampling noise causes the discriminator to move in different directions at each iteration, as a result of which the generator fluctuates. (iv) Even if the gradients of the discriminator point in the right direction, the learning rate of the generator might be too large. When this overshooting happens, the gradients of the discriminator lead in the wrong direction due to over-confidence (Mescheder et al., 2018).

---

[*] Corresponding Author: `yasin001@e.ntu.edu.sg`
[†] Joint last authors
[1] The code is available at `https://github.com/yasinyazici/EMA_GAN`

In this work, we explore in detail simple strategies for tackling the cycling behavior without influencing the adversarial game. Our strategies average generator parameters over time *outside* the training loop. Averaging generator and discriminator parameters is known to be an optimal solution for convex-concave min-max games (Freund & Schapire, 1999). However, no such guarantees are known (even for bilinear games) if we apply exponential discounting.

Our contributions are the following: (i) We show theoretically that although EMA does not converge to equilibrium, even in bilinear games, it nevertheless helps to stabilize cyclic behavior by shrinking its amplitude. In non-bilinear settings it preserves the stability of locally stable fixed points. (ii) We demonstrate that both averaging techniques consistently improve results for several different datasets, network architectures, and GAN objectives. (i) We compare it with several other methods that try to alleviate the cycling or non-convergence problem and demonstrate its unusual effectiveness.

## 2 RELATED WORK

There have been attempts to improve stability and alleviate cycling issues arising in GANs. Salimans et al. (2016) use historical averaging of both generator and discriminator parameters as a regularization term in the objective function: each model deviating from its time average is penalized to improve stability. Ge et al. (2018) update the discriminator by using a mixture of historical generators. Heusel et al. (2017) use two time scale update rules to converge to a local equilibrium.

Mescheder et al. (2017) identify the presence of eigenvalues of the Jacobian of the gradient vector with zero real part and large imaginary part as a reason for non-convergence. They address this issue by including squared gradients of the players to the objective. Extra-gradient (optimistic) methods help with convergence (Daskalakis et al., 2018; Mertikopoulos et al., 2019). Some of the above stated methods such as Salimans et al. (2016); Mescheder et al. (2017) influence the min-max game by introducing regularization terms into the objective function. Regularizing the adversarial objective can potentially change the optimal solution of the original objective. Our method on the other hand does not change the game dynamics, by keeping an average over iterates *outside* of the training loop.

**Averaging at maximum likelihood objective:** Uniform Average and Exponential Moving Average over model parameters has been used for a long time (Polyak & Juditsky, 1992; van den Oord et al., 2017; Athiwaratkun et al., 2018; Izmailov et al., 2018). However we would like to emphasize the difference between convex/non-convex optimization and saddle point optimization. GANs are an instance of the latter approach. There is no formal connection between the two approaches, so we cannot extrapolate our understanding of averaging at maximum likelihood objectives to saddle point objectives.

**Averaging at minimax objective:** Gidel et al. (2018) used uniform averaging similar to our work, however we show its shortcomings and how exponential decay averaging performs better. While this strategy has been used in (Karras et al., 2017) recently, their paper lacks any insights or a detailed analysis of this approach.

## 3 METHOD

In case of convex-concave min-max games, it is known that the average of generator/discriminator parameters is an optimal solution (Freund & Schapire, 1999), but there is no such guarantee for a non-convex/concave setting. Moving Average (MA) over parameters $\theta$ is an efficient implementation of uniform averaging without saving iterates at each time point:

$$\theta_{MA}^{(t)} = \frac{t-1}{t}\theta_{MA}^{(t-1)} + \frac{1}{t}\theta^{(t)} \tag{1}$$

When the generator iterates reaches a stationary distribution, Eq.1 approximates its mean. However there are two practical difficulties with this equation. First we do not know at which time point the iterates reach a stationary distribution. Second, and more importantly, iterates may not stay in the same distribution after a while. Averaging over samples coming from different distributions can produce worse results, which we have also observed in our own experiments.

Because of the issues stated above, we use Exponential Moving Average (EMA):

$$\theta_{EMA}^{(t)} = \beta\theta_{EMA}^{(t-1)} + (1-\beta)\theta^{(t)} \tag{2}$$

where $\theta_{EMA}^{(0)} = \theta^{(0)}$. EMA has an effective time window over iterates, where early iterates fade at a rate depending on the $\beta$ value. As $\beta$ approaches 1, averaging effectively computes longer time windows.

Both EMA and MA offer performance gains, with EMA typically being more robust (for more details see Section 5). Averaging methods are simple to implement and have minimal computation overhead. As they operate outside of the training loop, they do not influence optimal points of the game. Hyperparameter search for EMA can be done with a single training run by simultaneously keeping track of multiple averages with different $\beta$ parameters.

## 4 WHY DOES AVERAGING WORK?

### 4.1 GLOBAL ANALYSIS FOR SIMPLE BILINEAR GAMES

As we will show experimentally, parameter averaging provides significant benefits in terms of GAN training across a wide range of different setups. Although one cannot hope to completely justify this experimental success theoretically in general non-convex-concave settings, it is at least tempting to venture some informed conjectures about the root causes of this phenomenon. To do so, we focus on the simplest possible class of saddle point problems, the class of planar bilinear problems (i.e. zero-sum games such as Matching Pennies). This is a toy, albeit classic, pedagogical example used to explain phenomena arising in GANs. Specifically, similar to Goodfellow (2017), we also move to a continuous time model to simplify the mathematical analysis.

In terms of the moving average method, it is well known that time-averaging suffices to lead to convergence to equilibria (Freund & Schapire, 1999). In fact, it has recently been established that the smooth (continuous-time) analogues of first order methods such as online gradient descent (follow-the-regularized leader) in bilinear zero-sum games are recurrent (i.e. effectively periodic) with trajectories cycling back into themselves (Mertikopoulos et al., 2018). As a result, the time-average of the system converges fast, at a $\mathcal{O}(1/T)$ rate, to the solution of the saddle point problem (Mertikopoulos et al., 2018). These systems have connections to physics (Bailey & Piliouras, 2019).

In contrast, not much is known about the behavior of EMA methods even in the simplest case of zero-sum games. As we will show by analyzing the simplest case of a bilinear saddle problem $\max_x \min_y xy$, even in this toy case EMA does not lead to convergent behavior, but instead reduces the size of the oscillations, leading to more stable behavior whilst enabling exploration in a close neighborhood around the equilibrium. In this case the gradient descent dynamics are as follows:

$$\left\{ \begin{array}{ccc} \frac{dx}{dt} & = & y \\ \frac{dy}{dt} & = & -x \end{array} \right\} \Leftrightarrow \left\{ \begin{array}{ccc} \frac{dx}{dt} & = & \frac{\partial H}{\partial y} \\ \frac{dy}{dt} & = & -\frac{\partial H}{\partial x} \end{array} \right\}$$

This corresponds to a standard Hamiltonian system whose energy function is equal to $H = \frac{x^2+y^2}{2}$. All trajectories are cycles centered at $0$. In fact, the solution to the system has the form of a periodic orbit $(x(t), y(t)) = (a \cos(t), a \sin(t))$. These connections have recently been exploited by Balduzzi et al. (2018) to design training algorithms for GANs (see also Bailey & Piliouras (2019)). Here we take a different approach. For simplicity let's assume $a = 1$. The Exponential Moving Average method now corresponds to an integral of the form $C \cdot \int_0^T \beta^{T-t} \theta(t) dt$ where $\theta(t) = (\cos(t), \sin(t))$ and $C$ a normalizing term such that the EMA of a constant function is equal to that constant (i.e. $C = 1/\int_0^T \beta^{T-t} dt$). Some simple algebra implies that $C \cdot \int_0^T \beta^{T-t} \sin(t) dt = \frac{-\ln(\beta)}{1+\ln^2(\beta)} \frac{\beta^T - \cos(T) - \ln(\beta)\sin(T)}{1-\beta^T}$. As the terms $\beta^T$ vanish exponentially fast, this function reduces to $\frac{\ln(\beta)}{1+\ln^2(\beta)}(\cos(T) + \ln(\beta)\sin(T))$. This is a periodic function, similarly to the components of $\theta(t)$, but its amplitude is significantly reduced, and as a result it is more concentrated around its mean value (which is the same as its corresponding $\theta(t)$ component, i.e. zero). For $\beta$ very close to 1, e.g. $\beta = 0.999$ this periodic function can be further approximated by $\ln(\beta)\cos(T)$. In other words, no matter how close to 1 we choose the discounting factor $\beta$, even in the toy case of a planar bilinear saddle problem (i.e. the Matching Pennies game) EMA has a residual non-vanishing periodic component, alas of very small amplitude around the solution of the saddle point problem.

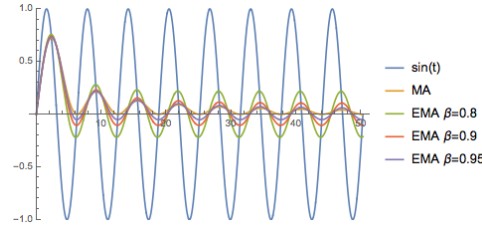

Figure 1: MA versus EMA for various values of $\beta$.

## 4.2 LOCAL STABILITY ANALYSIS FOR GANS

The model of the previous section introduced two simplifications so as to be theoretically tractable: i) it focused on a (small) bilinear game, and ii) it was a continuous time model. Despite these simplifications the global behavior was still rather complex, indicating the difficulty of completely analyzing a more realistic model. Nevertheless, in this section we examine such a model (discrete-time non-convex loss functions) and show that EMA preserves the stability of locally stable fixed points (Nash equilibria), providing further theoretical evidence in support of averaging. Due to the complexity of GANs, local stability analysis is often used as a method for obtaining theoretical insights, see e.g. Nagarajan & Kolter (2017); Mescheder et al. (2018).

Suppose we have a generator $G$ parameterized by $\theta \in \mathbf{R}^p$, and a discriminator $D$ parameterized by $\phi \in \mathbf{R}^q$; let $x = (\theta, \phi) \in \mathbf{R}^n$, $n = p + q$. Then, during training, $G$ and $D$ minimize loss functions $\mathcal{L}^{(\theta)}(\theta, \phi)$ and $\mathcal{L}^{(\phi)}(\theta, \phi)$ respectively. Following the same formulation as in Gidel et al. (2018):

$$\theta^* \in \operatorname*{argmin}_{\theta} \mathcal{L}^{(\theta)}(\theta, \phi^*) \qquad\qquad \phi^* \in \operatorname*{argmin}_{\phi} \mathcal{L}^{(\phi)}(\theta^*, \phi)$$

For specific possible instantiations of these loss functions see Goodfellow et al. (2014); Arjovsky et al. (2017). We define the gradient vector field $V(\theta, \phi) \in \mathbf{R}^n$ and its Jacobian $J_V(\theta, \phi) \in \mathbf{R}^{n \times n}$ as follows:

$$V(\theta, \phi) = \begin{bmatrix} \nabla_\theta \mathcal{L}^{(\theta)}(\theta, \phi) \\ \nabla_\phi \mathcal{L}^{(\phi)}(\theta, \phi) \end{bmatrix} \qquad J_V(\theta, \phi) = \begin{bmatrix} \nabla_\theta^2 \mathcal{L}^{(\theta)}(\theta, \phi) & \nabla_\phi \nabla_\theta \mathcal{L}^{(\theta)}(\theta, \phi) \\ \nabla_\theta \nabla_\phi \mathcal{L}^{(\phi)}(\theta, \phi) & \nabla_\phi^2 \mathcal{L}^{(\phi)}(\theta, \phi) \end{bmatrix}.$$

In this analysis, we assume that $G$ and $D$ are trained with some method that can be expressed as the iteration

$$x_t = F(x_{t-1}) \tag{3}$$

for some function $F$, where $x_t = (\theta_t, \phi_t)$ denotes the parameters obtained after $t$ training steps. This framework covers all training algorithms that rely only on a single past iterate to compute the next iterate, and includes simultaneous gradient descent and alternating gradient descent methods. More concretely, simultaneous gradient descent with a learning rate $\eta$ can be written in this form as follows

$$F_\eta(x) = x - \eta V(x). \tag{4}$$

EMA then averages the iterates $x_t$ to produce a new sequence of iterates $\hat{x}_t$ such that

$$
\begin{aligned}
\hat{x}_t &= \beta \hat{x}_{t-1} + (1 - \beta) x_t && \text{(by definition (2))} \\
&= \beta \hat{x}_{t-1} + (1 - \beta) F(x_{t-1}) && \text{(Replace } x_t \text{ using Eq. (3))} \\
&= \beta \hat{x}_{t-1} + (1 - \beta) F\left( \frac{\hat{x}_{t-1} - \beta \hat{x}_{t-2}}{1 - \beta} \right) && \text{(Replace } x_{t-1} \text{ using definition (2))}
\end{aligned}
$$

We have thus expressed the EMA algorithm purely in terms of a new set of iterates $\hat{x}_t$. Note that computing the next iterate now requires the past two iterates. Flammarion & Bach (2015) observed that the standard moving average can similarly be expressed as a second-order difference equation. We can then define the EMA operator $\hat{F}_\beta(x_t, x_{t-1})$ such that $\begin{bmatrix} x_t \\ x_{t-1} \end{bmatrix} = \hat{F}_\beta(x_{t-1}, x_{t-2})$

$$\hat{F}_\beta(x_t, x_{t-1}) = \begin{bmatrix} \beta \mathbf{I}_n & \mathbf{0}_n \\ \mathbf{I}_n & \mathbf{0}_n \end{bmatrix} \begin{bmatrix} x_t \\ x_{t-1} \end{bmatrix} + (1 - \beta) \begin{bmatrix} F(A[x_t \ x_{t-1}]^T) \\ \mathbf{0}_n \end{bmatrix}, \text{ for } A = \frac{1}{1 - \beta} [\mathbf{I}_n \ -\beta \mathbf{I}_n].$$

The Jacobian of $\hat{F}_\beta$, $J_{\hat{F}_\beta}$ is therefore

$$
\begin{aligned}
J_{\hat{F}_\beta}(x_t, x_{t-1}) &= \begin{bmatrix} \beta\mathbf{I}_n & \mathbf{0}_n \\ \mathbf{I}_n & \mathbf{0}_n \end{bmatrix} + (1-\beta)\begin{bmatrix} J_F(A[x_t \ x_{t-1}]^T) \cdot A \\ \mathbf{0}_n \end{bmatrix} \\
&= \begin{bmatrix} \beta\mathbf{I}_n + J_F(A[x_t \ x_{t-1}]^T) & -\beta J_F(A[x_t \ x_{t-1}]^T) \\ \mathbf{I}_n & \mathbf{0}_n \end{bmatrix}
\end{aligned}
$$

We can now proceed to explore the behavior of the EMA algorithm near a local Nash equilibrium by examining the eigenvalues of its Jacobian $J_{\hat{F}_\beta}$, which are described by Theorem 1.

**Theorem 1.** *The eigenvalues of the Jacobian $J_{\hat{F}_\beta}(x^*, x^*)$ at a local Nash equilibrium $x^*$ are $\beta$ (with multiplicity $n$) and the eigenvalues of $J_F(x^*)$, the Jacobian of the training operator at the equilibrium.*

*Proof.* We solve $\det(\gamma\mathbf{I}_{2n} - J_{\hat{F}_\beta}) = 0$ to obtain eigenvalues $\gamma$. Some algebra reveals that

$$
\begin{aligned}
\det(\gamma\mathbf{I}_{2n} - J_{\hat{F}_\beta}) &= \begin{vmatrix} (\gamma-\beta)\mathbf{I}_n - J_F(x^*) & \beta J_F(x^*) \\ -\mathbf{I}_n & \gamma\mathbf{I}_n \end{vmatrix} \\
&= |\gamma(\gamma-\beta)\mathbf{I}_n - \gamma J_F(x^*) + \beta J_F(x^*)| \\
&= |(\gamma-\beta)(\gamma\mathbf{I}_n - J_F(x^*))| \\
&= (\gamma-\beta)^n \det(\gamma\mathbf{I}_n - J_F(x^*))
\end{aligned}
$$

where we used the following equality (e.g. see Lemma 1 in Gidel et al. (2018)): $\left|\begin{smallmatrix} A & B \\ C & D \end{smallmatrix}\right| = |AD - BC|$ if $C$ and $D$ commute. Thus, $\det(\gamma\mathbf{I}_{2n} - J_{\hat{F}_\beta}) = 0$ if $\gamma = \beta$ or $\det(\gamma\mathbf{I}_n - J_F(x^*)) = 0$ so $\gamma$ is an eigenvalue of $J_F(x^*)$. □

Given that $0 < \beta < 1$ in the EMA algorithm, we have the following corollary that formalizes the intuition that the dynamics of EMA near a local Nash equilibrium are tightly linked to those of the underlying training algorithm.

**Corollary 1.1.** *Assuming that $0 < \beta < 1$, the eigenvalues of $J_F$ lie within the unit ball if and only if the eigenvalues of $J_{\hat{F}_\beta}$ lie within the unit ball.*

Also, we see that EMA could potentially slow down the rate of convergence compared to the underlying training method if $\beta$ is larger than the magnitude of all other eigenvectors of $J_{F_\beta}$.

## 5 EXPERIMENTS

We use both illustrative examples (i.e. mixtures of Gaussians) as well as four commonly used real-world datasets, namely CIFAR-10 (Krizhevsky et al., 2009), STL-10 (Coates et al., 2011), CelebA (Liu et al., 2015), and ImageNet (Russakovsky et al., 2015) to show the effectiveness of averaging. For mixtures of Gaussians, the architecture, hyperparameters and other settings are defined in its own section. For CelebA, 64x64 pixel resolution is used, while all other experiments are conducted on 32x32 pixel images.

All dataset samples are scaled to $[-1, 1]$ range. No labels are used in any of the experiments. For the optimizer, we use ADAM (Kingma & Ba, 2014) with $\alpha = 0.0002$, $\beta_1 = 0.0$ and $\beta_2 = 0.9$. For the GAN objective, we use the original GAN (Referring to Non-Saturating variant) (Goodfellow et al., 2014) objective and the Wasserstein-1 (Arjovsky et al., 2017) objective, with Lipschitz constraint satisfied by gradient penalty (Gulrajani et al., 2017). Spectral normalization (Miyato et al., 2018) is only used with the original objective. In all experiments, the updates are alternating gradients.

The network architecture is similar to a full version of a progressively growing GAN (Karras et al., 2017), with details provided in the Appendix. We call this architecture conventional. ResNet from Gulrajani et al. (2017) is used as a secondary architecture. Unless stated otherwise, the objective is the original one, the architecture is conventional, discriminator to generator update ratio is 1, $\beta$ value is 0.9999 and MA starting point is 100k. Inception (Salimans et al., 2016) and FID (Heusel et al., 2017) scores are used as quantitative measures. Higher inception scores and lower FID scores

are better. We have used the online Chainer implementation[2] for both scores. For Inception score evaluation, we follow the guideline from Salimans et al. (2016). The FID score is evaluated on 10k generated images and statistics of data are calculated at the same scale of generation e.g. 32x32 data statistics for 32x32 generation. The maximum number of iterations for any experiment is 500k. For each experiment, we state the training duration. At every 10k iterations, Inception and FID scores are evaluated, and we pick the best point. All experiments are repeated at least 3 times with random initializations to show that the results are consistent across different runs.

In addition to these experiments, we also compare the averaging methods, with *Consensus Optimization* (Mescheder et al., 2017), *Optimistic Adam* (Daskalakis et al., 2018) and *Zero Centered Gradient Penalty* (Mescheder et al., 2018) on the Mixture of Gaussians data set and CIFAR-10. For CIFAR-10 comparison, we use a DCGAN-like architecture Radford et al. (2015) and follow the same hyperparameter setting as in Mixture of Gaussians except WGAN-GP's $\lambda$ which is 10.0.

To ensure a fair comparison, we use the same samples from the prior distribution for both averaged and non-averaged generators. In this way, the effect of averaging can be directly observed, as the generator generates similar images. We did not cherry-pick any images: all images generated are from randomly selected samples from the prior. For extended results refer to the Appendix.

## 5.1 MIXTURE OF GAUSSIANS

This illustrative example is a two-dimensional mixture of 16 Gaussians, where the mean of each Gaussian lies on the intersection points of a 4x4 grid. Each Gaussian is isotropic with $\sigma = 0.2$. Original GAN (Goodfellow et al., 2014) is used as the objective function with Gradient Penalty from Gulrajani et al. (2017), with 1:1 discriminator/generator update and alternative update rule. We run the experiment for 40k iterations, and measure Wasserstein-1[3] distance with 2048 samples from both distributions at every 2k iterations. We take the average of the last 10 measurements from the same experiment and repeat it 5 times. We compare this baseline with *Optimistic Adam* (OMD), *Consensus Optimization* (CO), and *Zero Centered Gradient Penalty* (Zero-GP) by using the same architecture, and following the original implementations as closely as possible. We have also combined EMA and MA with OMD, CO and Zero-GP to see whether they can improve over these methods. Detailed settings are listed in the Appendix.

Table 1 shows the distance for various methods. In all cases, EMA outperforms other methods, and interestingly, it also improves OMD, CO and Zero-GP. This indicates that OMD, CO and Zero-GP do not necessarily converge in non-convex/concave settings, but still cycle. MA also improves the results in certain cases, but not as much as EMA. Our observation is that uniform averaging over long time iterates in the non-convex/concave case hurts performance (more on this in later experiments). Because of this, we have started uniform averaging at later stages in training and treat it as a hyper-parameter. For this experiment, the start interval is selected as 20k.

Table 1: Wasserstein-1 Distance for non-averaged generator, EMA, MA, *Optimistic Adam*, *Consensus Optimization* and Zero-GP

|  | no-average | EMA | MA |
| --- | --- | --- | --- |
| Baseline | $0.0419 \pm 0.0054$ | $0.0251 \pm 0.0026$ | $0.0274 \pm 0.0016$ |
| *Optimistic Adam* | $0.0494 \pm 0.0041$ | $0.0431 \pm 0.0042$ | $0.0525 \pm 0.0024$ |
| *Consensus Optimization* | $0.0286 \pm 0.0034$ | $0.0252 \pm 0.0035$ | $0.0390 \pm 0.0105$ |
| *Zero-GP* | $0.0363 \pm 0.0018$ | $0.0246 \pm 0.0068$ | $0.0317 \pm 0.0121$ |

Figure 2 is an illustration of the above results. The blue regions indicate a lack of support, while the red ones indicate a region of support. Without averaging, the generator omits certain modes at different time points, and covers them later while omitting previously covered modes. We observe a clear fluctuation in this case, while the EMA consistently shows more balanced and stable supports

---

[2] https://github.com/pfnet-research/chainer-gan-lib/blob/master/common/evaluation.py

[3] Python Optimal Transport package at http://pot.readthedocs.io/en/stable/

across different iterations. OMD, CO and Zero-GP perform better than the baseline, however they seem to be less stable across different iterations than EMA results.

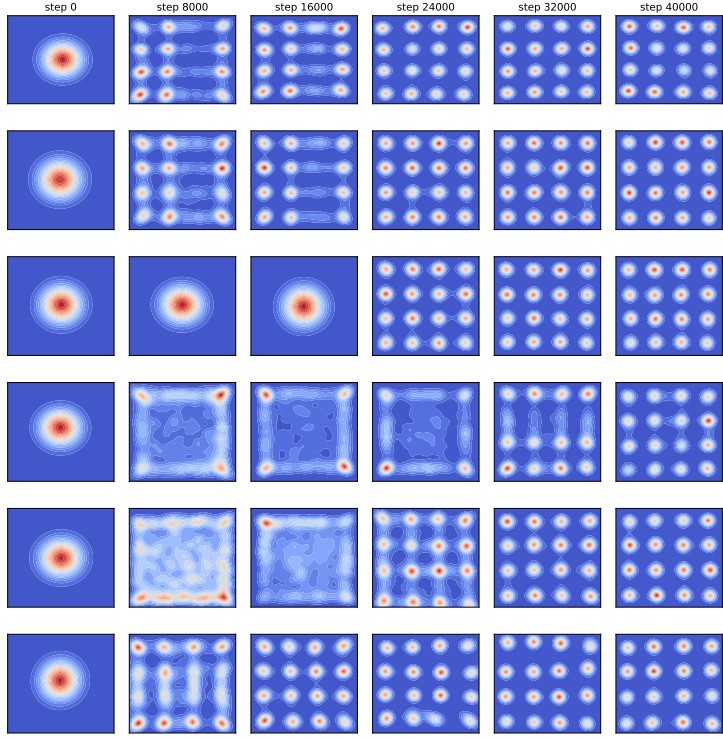

Figure 2: Effect of averaging parameters for Mixture of Gaussians. From top to bottom: without average, EMA, MA, *Optimistic Adam*, *Consensus Optimization*, *Zero-GP*.

Table 2: Inception and FID scores on CIFAR-10, STL-10 and ImageNet. (100k), (250k) etc. refer to number of iterations of the generator. Experiments were repeated 3 times.

| | IS | | | FID | | |
|---|---|---|---|---|---|---|
| | no average | EMA | MA | no average | EMA | MA |
| **CIFAR-10** | | | | | | |
| conventional (500k) | $8.14 \pm 0.01$ | $8.89 \pm 0.12$ | $8.74 \pm 0.19$ | $16.84 \pm 1.00$ | $12.56 \pm 0.17$ | $16.59 \pm 0.52$ |
| conventional WGAN-GP $n_{dis} = 1$ (250k) | $7.18 \pm 0.06$ | $7.85 \pm 0.08$ | $7.93 \pm 0.15$ | $25.51 \pm 1.22$ | $19.18 \pm 0.95$ | $22.71 \pm 0.95$ |
| conventional WGAN-GP $n_{dis} = 5$ (250k) | $7.22 \pm 0.07$ | $7.92 \pm 0.10$ | $8.20 \pm 0.06$ | $25.32 \pm 0.70$ | $19.72 \pm 0.31$ | $20.02 \pm 0.75$ |
| ResNet $n_{dis} = 5$ (120k) | $7.86 \pm 0.13$ | $8.46 \pm 0.13$ | $8.68 \pm 0.09$ | $20.64 \pm 1.38$ | $17.58 \pm 1.84$ | $18.30 \pm 1.32$ |
| ResNet WGAN-GP $n_{dis} = 1$ (250k) | $7.63 \pm 0.12$ | $8.22 \pm 0.05$ | $8.51 \pm 0.07$ | $21.49 \pm 0.40$ | $15.85 \pm 0.27$ | $16.27 \pm 0.56$ |
| ResNet WGAN-GP $n_{dis} = 5$ (250k) | $7.38 \pm 0.26$ | $7.94 \pm 0.31$ | $8.32 \pm 0.25$ | $23.20 \pm 2.62$ | $19.41 \pm 2.57$ | $19.33 \pm 3.17$ |
| **STL-10** | | | | | | |
| conventional (250k) | $7.96 \pm 0.35$ | $8.39 \pm 0.10$ | $8.23 \pm 0.24$ | $22.33 \pm 1.59$ | $19.64 \pm 1.38$ | $26.11 \pm 2.27$ |
| **ImageNet** | | | | | | |
| conventional (500k) | $8.33 \pm 0.10$ | $8.88 \pm 0.14$ | $8.33 \pm 0.03$ | $24.59 \pm 0.59$ | $21.83 \pm 0.93$ | $26.77 \pm 1.12$ |

## 5.2 QUALITATIVE SCORES

Table 2 tabulates Inception and FID scores with no averaging, EMA and MA. In all datasets, significant improvements are seen in both scores for EMA. EMA achieves better results more consistently on different datasets when compared to MA. The main reason why MA gets worse results is that it averages over longer iterates with equal weights (refer to Figure 4 and others in the Appendix). We have observed improvements when the starting point of MA is pushed later in training, however for convenience and ease of experimentation our main results are based on the EMA method.

## 5.3 CIFAR-10

Figure 3 shows images generated with and without EMA for CIFAR-10 after 300k iterations. Objects exhibit fewer artifacts and look visually better with EMA. It is interesting to note that the averaged and non-averaged versions converge to the same object, but can have different color and texture. Figure 4 shows Inception and FID scores during training. EMA model outperform its non-averaged counter-part consistently by a large margin, which is more or less stable throughout the training. However MA model's FID performance reduces gradually as it considers longer time windows for averaging. This observation also holds in STL10 and ImageNet experiments (See Appendix). Our intuition is that this phenomenon occurs because generator iterates does not stay in the same parameter region and averaging over parameters of different regions produces worse results. Besides, we have seen clear disagreement between FID and IS in case of MA. We think this happens because IS is not a proper metric to measure difference between two distributions and has many known flaws such as it does not measure intra-class diversity (Barratt & Sharma, 2018; Zhou et al., 2018; Heusel et al., 2017).

Figure 5 compares EMA, CO and OMD. EMA clearly improves baseline and OMD in both scores, while its improvement on CO is smaller but still consistent.

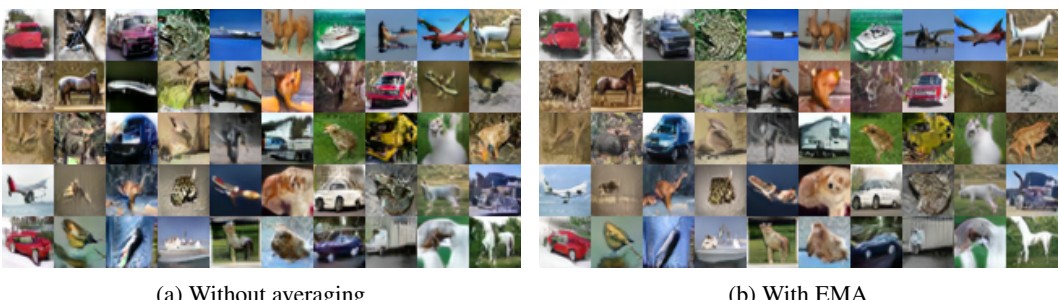

(a) Without averaging (b) With EMA

Figure 3: Generation for CIFAR-10 dataset.

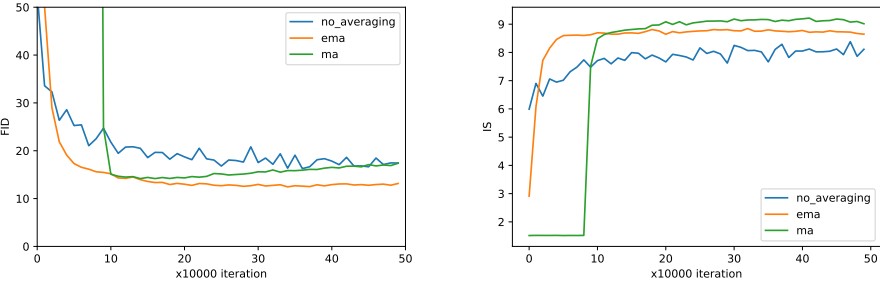

Figure 4: CIFAR-10 FID and IS scores during training. Setting: Original GAN objective, conventional architecture, $n_{dis} = 1$

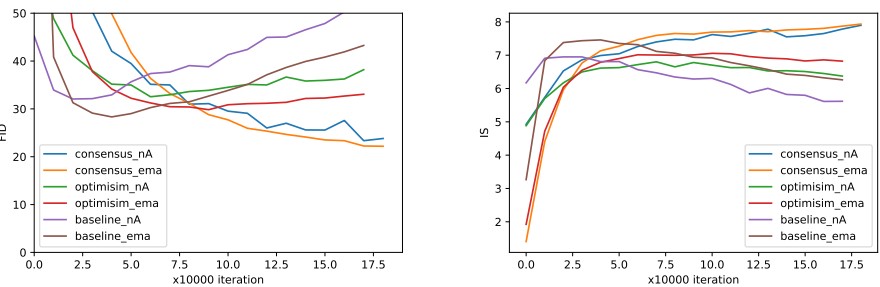

Figure 5: Comparison of EMA, *Optimistic Adam* and *Consensus Optimization* on CIFAR-10.

## 5.4 CELEBA

Figure 6 shows the same 10 samples from the prior fed to the non-averaged generator, the EMA generator with various $\beta$ values, and the MA generator at 250k iterations in the training. The non-averaged images show strong artifacts, but as we average over longer time windows, image quality improves noticeably. Interestingly, significant attribute changes occurs as $\beta$ approaches 1. There is a shift in gender, appearance, hair color, background etc., although images are still similar in pose and composition. EMA results with $\beta = 0.999$ and $\beta = 0.9999$ also look better than MA results, which are averaged over the last 200k iterates. This shows there is an optimal $\beta$/window size to tune for best results.

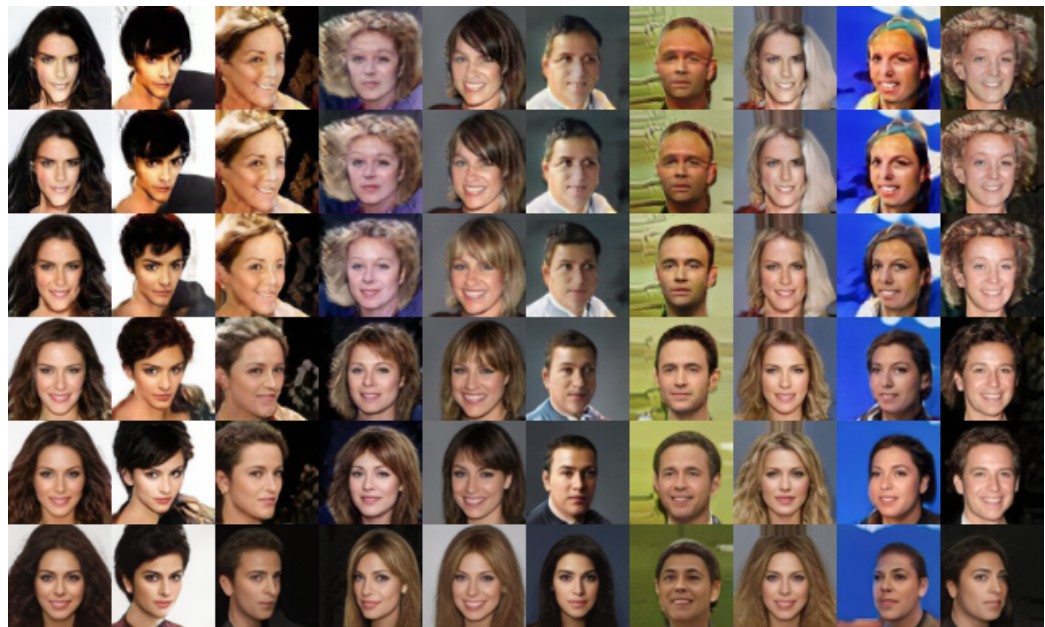

Figure 6: Generation for CelebA dataset for various $\beta$ values at 250k iterations. From top to bottom: (a) non-averaged generator, (b) $\beta = 0.9$, (c) $\beta = 0.99$, (d) $\beta = 0.999$, (e) $\beta = 0.9999$, (f) MA.

Figure 7 compares the stability of a non-averaged generator and EMA by using the same 2 samples from the prior. Starting from 50k iteration, images are generated from both models with 10k interval until 200k iterations. Images from the non-averaged generator change attributes frequently, while only preserving generic image composition. Meanwhile, the averaged generator produces smoother changes, with attributes changing more slowly.

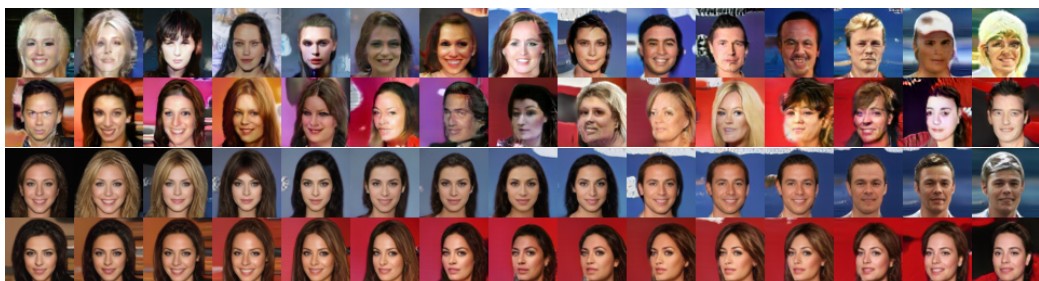

Figure 7: Generation for CelebA dataset for the same 2 noise samples from 50k to 200k with 10k intervals. (top 2 rows) without averaging, (bottom two rows) with EMA $\beta = 0.9999$

## 5.5 STL-10 & IMAGENET

Figure 8 shows images generated with and w/o EMA for STL-10 and ImageNet. Even though quantitative scores are better for EMA, we do not see as clear visual improvements as in previous results but rather small changes. Both models produce images that are unrecognizable to a large degree. This observation strengthens our intuition that averaging brings the cycling generator closer to the local optimal point, but it does not necessarily find a good solution as the local optimum may not be good enough.

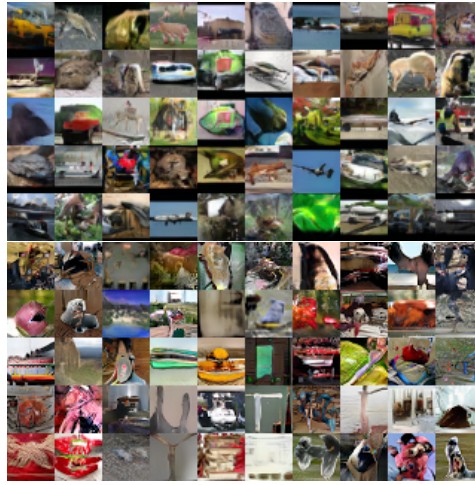 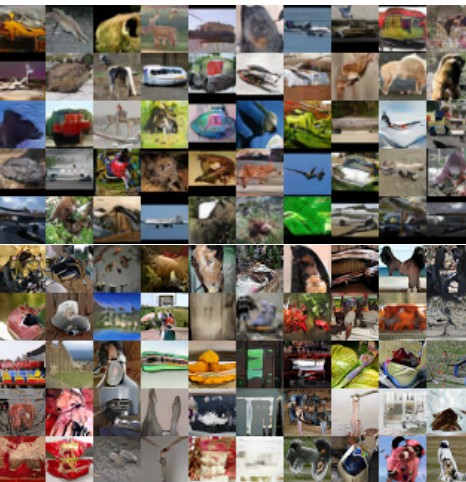

Figure 8: Generation for STL-10 & ImageNet dataset after 500k iteration. (Top): STL-10, (Bottom):ImageNet, (Left): w/o averaging (Right): with EMA

## 6 CONCLUSION

We have explored the effect of two different techniques for averaging parameters outside of the GAN training loop, moving average (MA) and exponential moving average (EMA). We have shown that both techniques significantly improve the quality of generated images on various datasets, network architectures and GAN objectives. In the case of the EMA technique, we have provided the first theoretical analysis of its implications, showing that even in simple bilinear settings it converges to stable limit cycles of small amplitude around the solution of the saddle problem. Averaging methods are easy to implement and have minimal computation overhead. As a result, these techniques are readily applicable in a wide range of settings. In the future, we plan to explore their effect on larger scales as well as on conditional GANs.

ACKNOWLEDGMENTS

Yasin Yazıcı was supported by a SINGA scholarship from the Agency for Science, Technology and Research (A*STAR). Georgios Piliouras would like to acknowledge SUTD grant SRG ESD 2015 097, MOE AcRF Tier 2 grant 2016-T2-1-170, grant PIE-SGP-AI-2018-01 and NRF 2018 Fellowship NRF-NRFF2018-07. This research was carried out at Advanced Digital Sciences Center (ADSC), Illinois at Singapore Pt Ltd, Institute for Infocomm Research (I2R) and at the Rapid-Rich Object Search (ROSE) Lab at the Nanyang Technological University, Singapore. The ROSE Lab is supported by the National Research Foundation, Singapore, and the Infocomm Media Development Authority, Singapore. Research at I2R was partially supported by A*STAR SERC Strategic Funding (A1718g0045). The computational work for this article was partially performed on resources of the National Supercomputing Centre, Singapore (https://www.nscc.sg). We would like to thank Houssam Zenati and Bruno Lecouat for sharing illustrative example codes.

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

## A ADDITIONAL RESULTS AND EXPERIMENTS

### A.1 PARAMETERIZED ONLINE AVERAGING

As an additional method we have used parameterized online averaging:

$$\theta_{MA}^{(t)} = \frac{t - \alpha}{t} \theta_{MA}^{(t-1)} + \frac{\alpha}{t} \theta^{(t)} \tag{5}$$

where $\alpha$ regulates the importance of the terms. When $\alpha = 1$, the equation corresponds to moving average (Eq. 1). As $\frac{t-\alpha}{t}$ might be negative, this algorithm should be restricted to the case of $t \geq \alpha$. We have started to apply it when $t \geq \alpha$. We have experimented (Table 3) with this method on CIFAR-10 with the same setting of the first row of Table 2. $\alpha = 1$ has failed as it started averaging from the first iteration which aligns with out earlier observations. $\alpha = 10, 100, 1000$ improves the scores at varying degree, however none is as good as EMA.

Table 3: IS and FID on CIFAR-10 (the same setting of the first row of Table 2) with various $\alpha$ values for parameterized online averaging.

|                    | IS               | FID               |
| ------------------ | ---------------- | ----------------- |
| nA                 | $8.02 \pm 0.20$  | $17.32 \pm 0.53$  |
| $\alpha = 1$       | failed           | failed            |
| $\alpha = 10$      | $8.58 \pm 0.06$  | $14.55 \pm 1.01$  |
| $\alpha = 100$     | $8.48 \pm 0.06$  | $13.85 \pm 0.51$  |
| $\alpha = 1000$    | $8.28 \pm 0.04$  | $14.64 \pm 0.51$  |
| $\alpha = 10000$   | $8.11 \pm 0.13$  | $15.60 \pm 0.59$  |

### A.2 QUANTITATIVE EFFECTS OF DIFFERENT $\beta$ FOR EMA

In the main body of this paper, we have used $\beta = 0.9999$ in all experiments but CelebA as it performs the best. In this section we show the effect of $\beta$ to quantitative scores. We have experimented different $\beta$ values ranging from 0.9 to 0.9999 on CIFAR-10 with the same setting of the first row of Table 2. We have run the experiment four times and provide the mean and standard deviation in Table 4. Each experiment runs for 500k. IS, FID score are collected with 20k iteration interval and best scores are published. From Table 4, we can see clear and progressive improvement when $\beta$ increases from 0.9 to 0.9999 in both IS and FID.

Table 4: IS and FID on CIFAR-10 (the same setting of the first row of Table 2) with various $\beta$ values for EMA.

|                        | IS              | FID              |
| ---------------------- | --------------- | ---------------- |
| nA                     | $8.09 \pm 0.07$ | $18.14 \pm 1.08$ |
| EMA $\beta = 0.9$      | $8.19 \pm 0.09$ | $17.82 \pm 0.43$ |
| EMA $\beta = 0.99$     | $8.32 \pm 0.03$ | $15.64 \pm 0.58$ |
| EMA $\beta = 0.999$    | $8.69 \pm 0.11$ | $13.77 \pm 0.38$ |
| EMA $\beta = 0.9999$   | $9.05 \pm 0.09$ | $12.91 \pm 0.49$ |

## B    Further Results

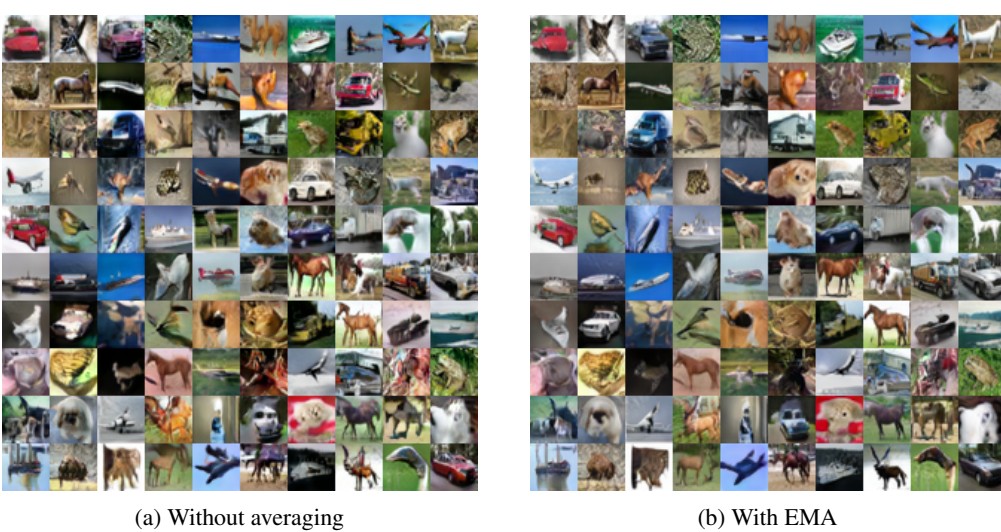

(a) Without averaging                                    (b) With EMA

Figure 9: Generation for CIFAR-10 dataset after 300k iterations.

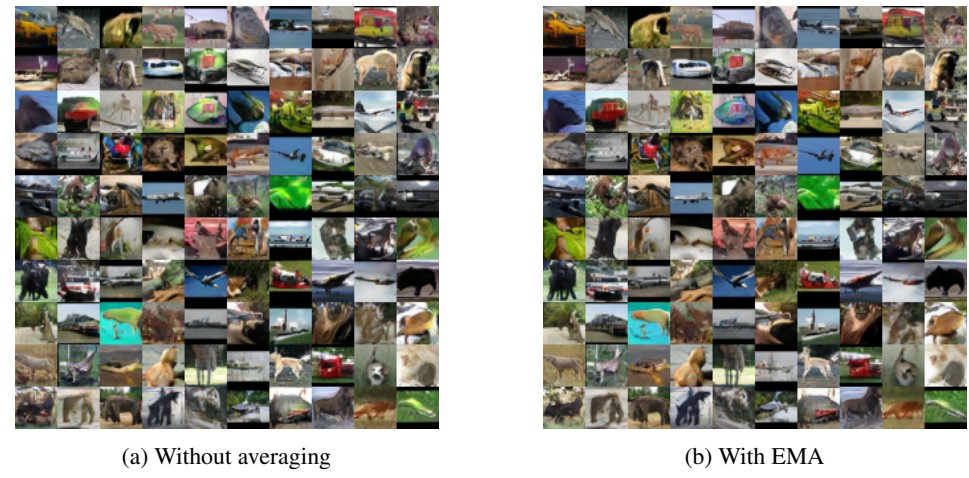

(a) Without averaging                                    (b) With EMA

Figure 10: Generation for STL-10 dataset after 500k iterations.

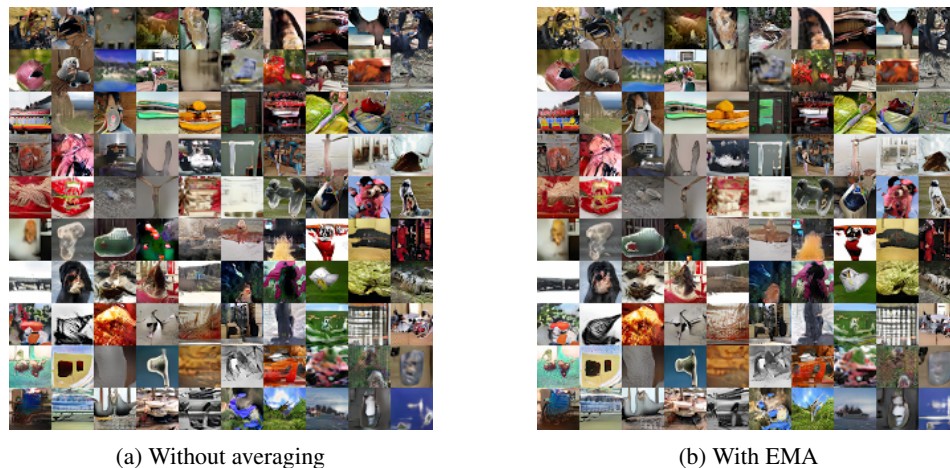

(a) Without averaging           (b) With EMA

Figure 11: Generation for ImageNet dataset after 500k iterations.

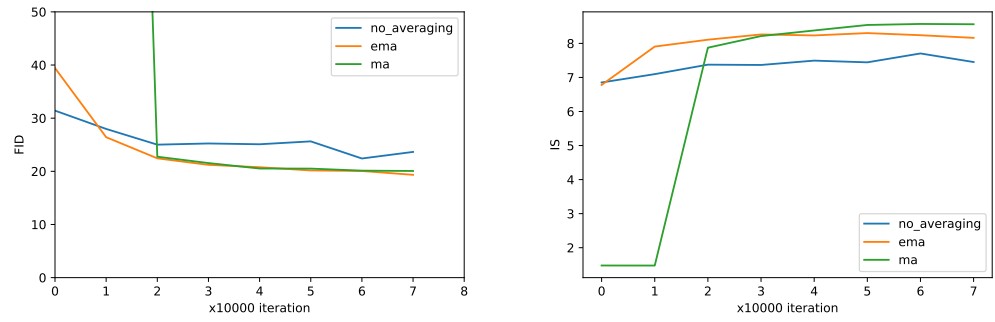

Figure 12: CIFAR-10 FID and IS scores during training. Setting: Original GAN objective/ ResNet Architecture/ $n_{dis} = 5$

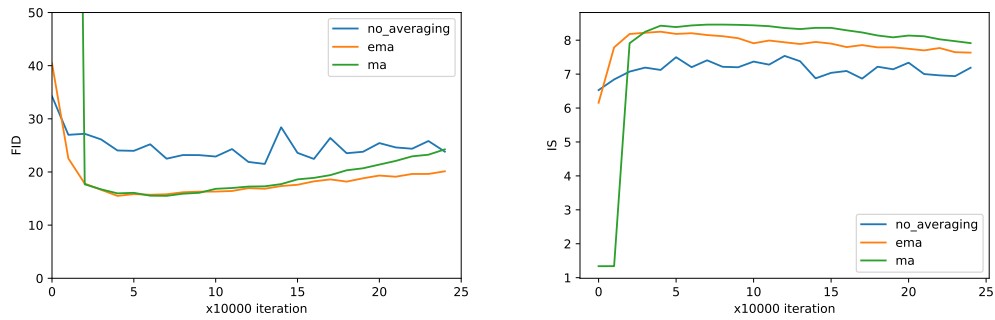

Figure 13: CIFAR-10 FID and IS scores during training. Setting: WGAN-GP objective/ ResNet Architecture/ $n_{dis} = 1$

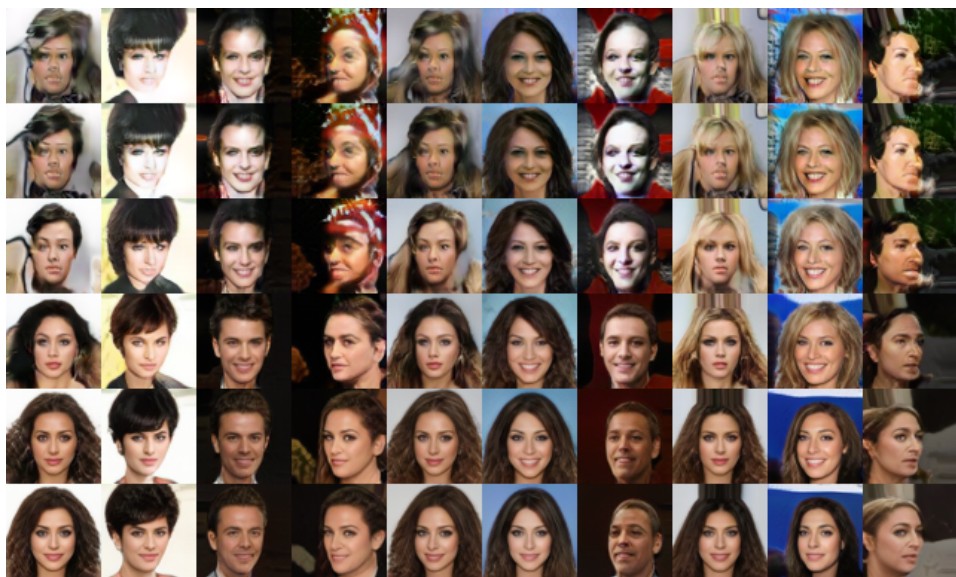

Figure 14: Generation for CelebA dataset for various $\beta$ values at 100k iteration. From top to bottom rows: (a) non-averaged generator, (b) $\beta = 0.9$, (c) $\beta = 0.99$, (d) $\beta = 0.999$, (e) $\beta = 0.9999$, (f) MA

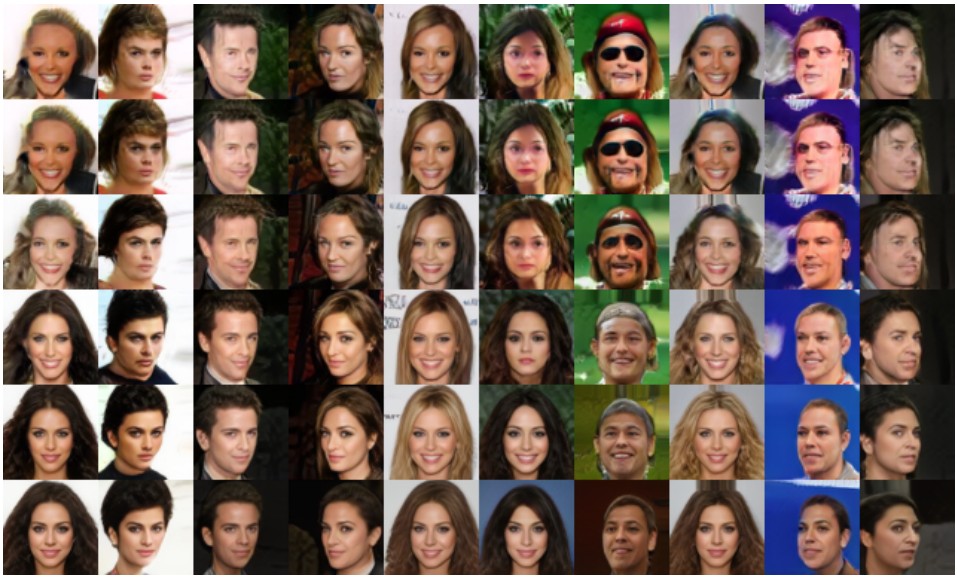

Figure 15: Generation for CelebA dataset for various $\beta$ values at 150k iteration. From top to bottom rows: (a) non-averaged generator, (b) $\beta = 0.9$, (c) $\beta = 0.99$, (d) $\beta = 0.999$, (e) $\beta = 0.9999$, (f) MA

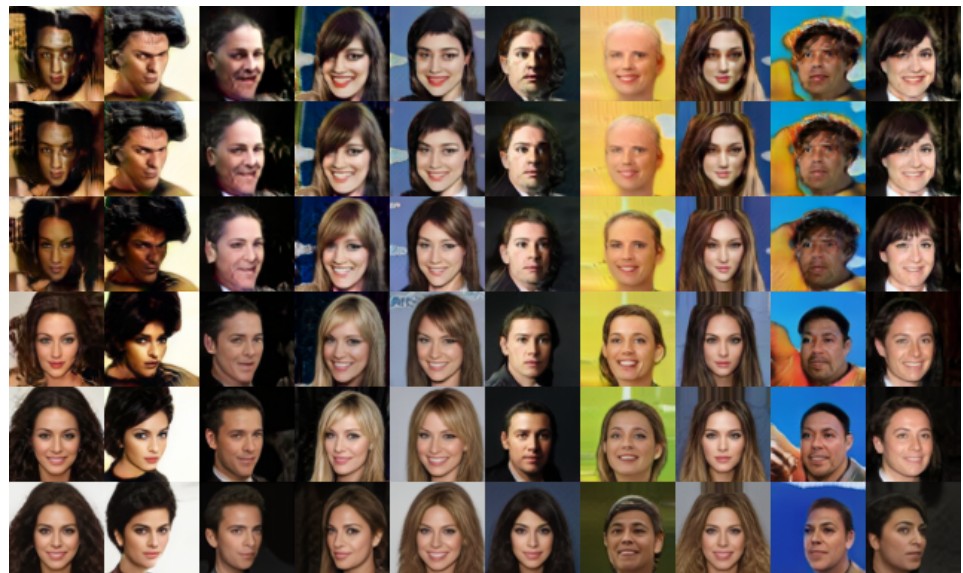

Figure 16: Generation for CelebA dataset for various $\beta$ values at 200k iteration. From top to bottom rows: (a) non-averaged generator, (b) $\beta = 0.9$, (c) $\beta = 0.99$, (d) $\beta = 0.999$, (e) $\beta = 0.9999$, (f) MA

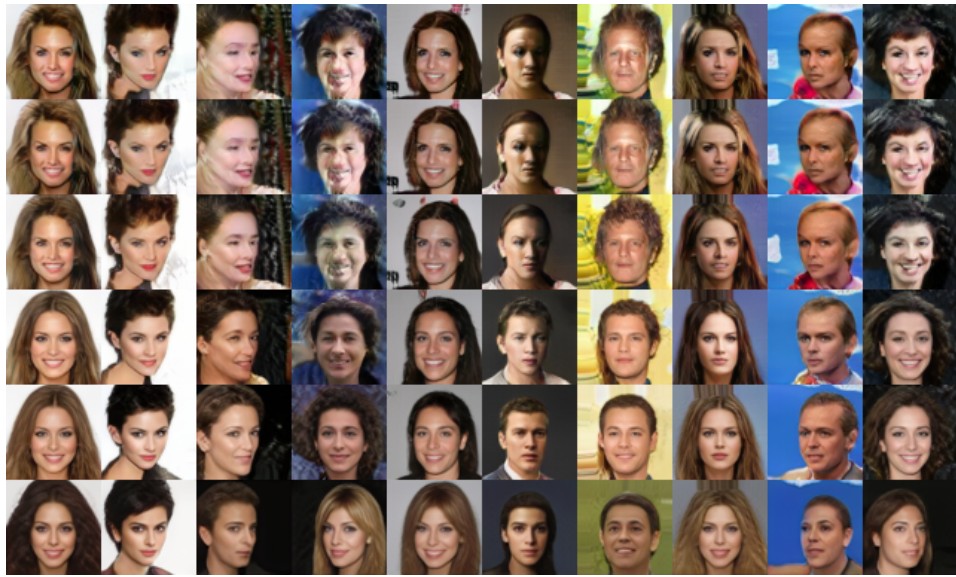

Figure 17: Generation for CelebA dataset for various $\beta$ values at 300k iteration. From top to bottom rows: (a) non-averaged generator, (b) $\beta = 0.9$, (c) $\beta = 0.99$, (d) $\beta = 0.999$, (e) $\beta = 0.9999$, (f) MA

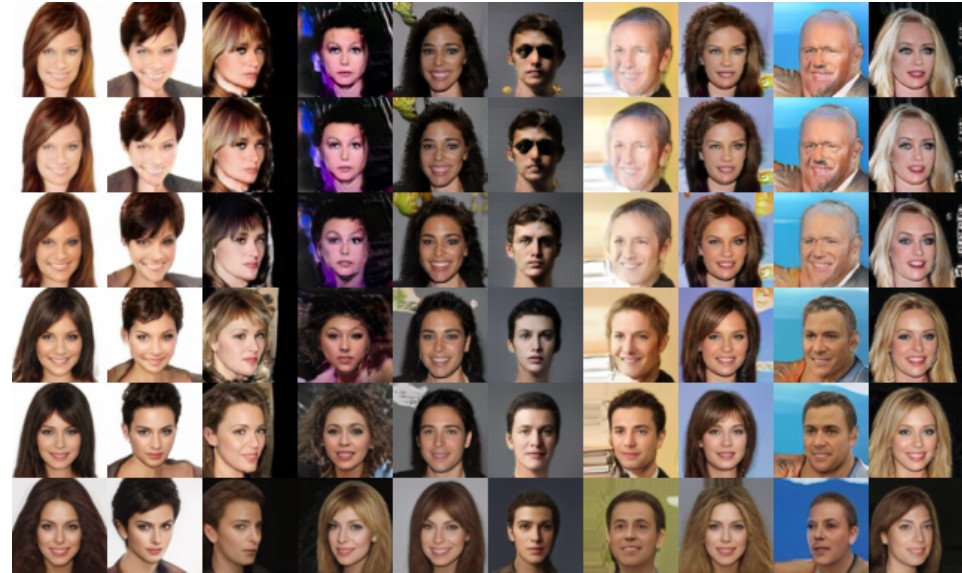

Figure 18: Generation for CelebA dataset for various $\beta$ values at 350k iteration. From top to bottom rows: (a) non-averaged generator, (b) $\beta = 0.9$, (c) $\beta = 0.99$, (d) $\beta = 0.999$, (e) $\beta = 0.9999$, (f) MA

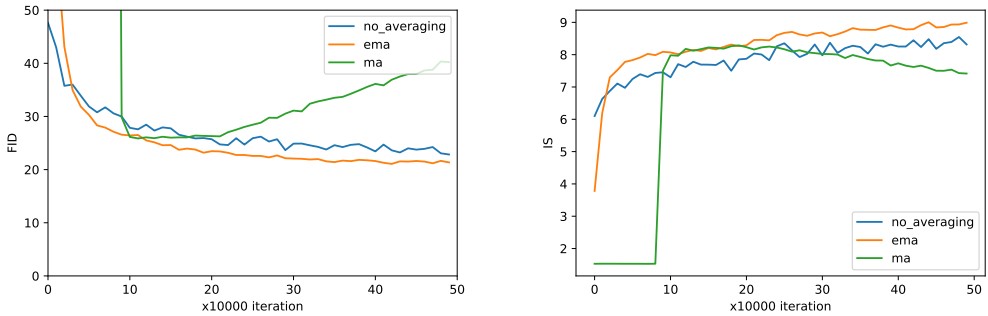

Figure 19: ImageNet FID and IS score during training. Setting: Original GAN objective/ Conventional Architecture/ $n_{dis} = 1$

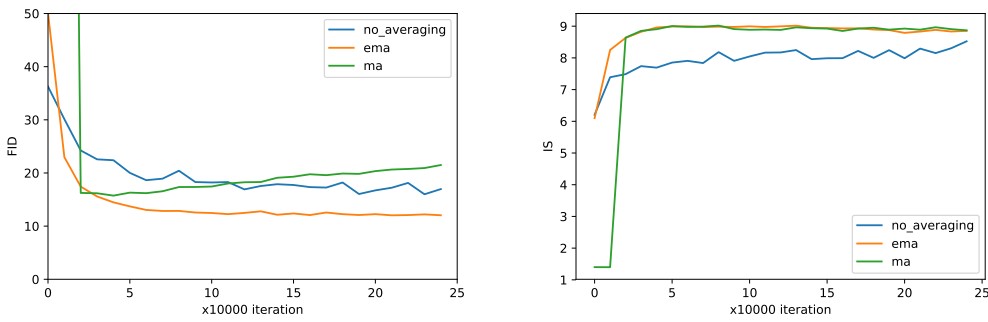

Figure 20: CIFAR-10 FID and IS scores during training. Setting: Original GAN objective/ Conventional Architecture/ $n_{dis} = 1$

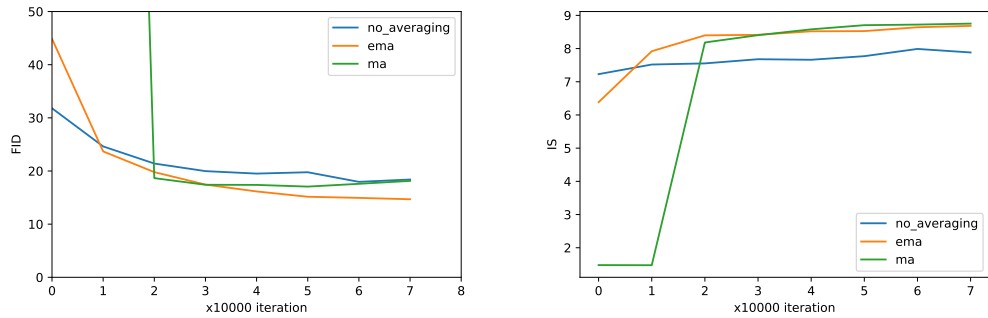

Figure 21: CIFAR-10 FID and IS scores during training. Setting: Original GAN objective/ Conventional Architecture/ $n_{dis} = 5$

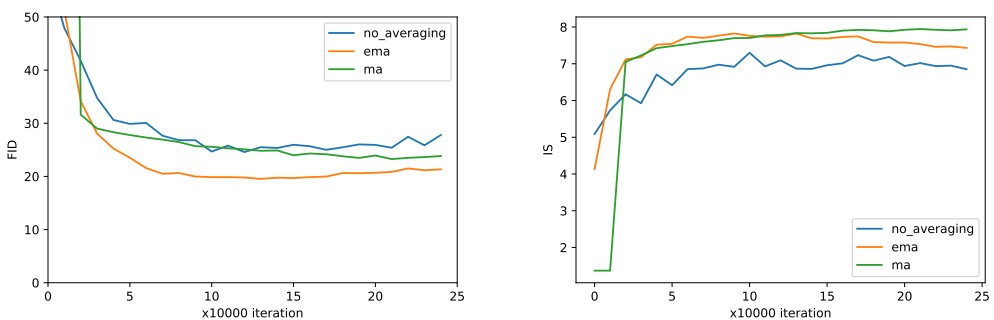

Figure 22: CIFAR-10 FID and IS scores during training. Setting: WGAN-GP objective/ Conventional Architecture/ $n_{dis} = 1$

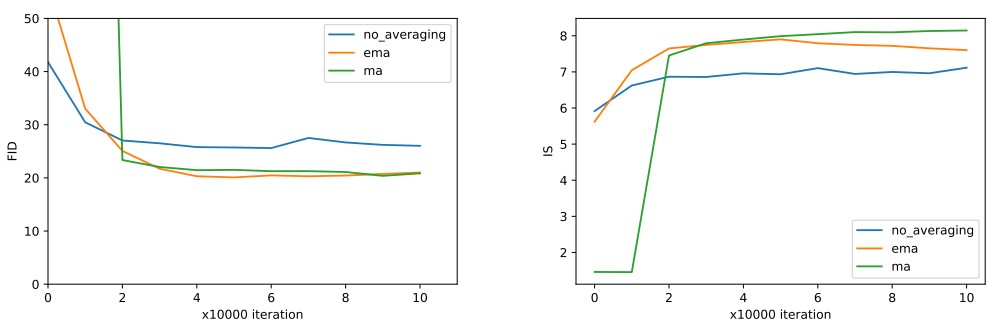

Figure 23: CIFAR-10 FID and IS scores during training. Setting: WGAN-GP objective/ Conventional Architecture/ $n_{dis} = 5$

## C  NETWORK ARCHITECTURES

When spectral normalization is used in ResNet, the feature number in each layer doubled by 2. Prior distribution for the generator is a 512-dimensional isotropic Gaussian distribution for conventional architecture and 128 for ResNet. Samples from the distribution are normalized to make them lie on a unit hypersphere before passing them into the generator.

Table 5: Conventional Generator Architecture for 32x32 resolution

| Layers | Act. | Output Shape |
|---|---|---|
| Latent vector | - | 512 x 1 x 1 |
| Conv 4 x 4 | BatchNorm - LReLU | 512 x 4 x 4 |
| Conv 3 x 3 | BatchNorm - LReLU | 512 x 4 x 4 |
| Upsample | - | 512 x 8 x 8 |
| Conv 3 x 3 | BatchNorm - LReLU | 256 x 8 x 8 |
| Conv 3 x 3 | BatchNorm - LReLU | 256 x 8 x 8 |
| Upsample | - | 256 x 16 x 16 |
| Conv 3 x 3 | BatchNorm - LReLU | 128 x 16 x 16 |
| Conv 3 x 3 | BatchNorm - LReLU | 128 x 16 x 16 |
| Upsample | - | 128 x 32 x 32 |
| Conv 3 x 3 | BatchNorm - LReLU | 64 x 32 x 32 |
| Conv 3 x 3 | BatchNorm - LReLU | 64 x 32 x 32 |
| Conv 1 x 1 | - | 3 x 32 x 32 |

Table 6: Conventional Discriminator Architecture for 32x32 resolution

| Layers | Act. | Output Shape |
|---|---|---|
| Input image | - | 3 x 32 x 32 |
| Conv 1 x 1 | LReLU | 64 x 32 x 32 |
| Conv 3 x 3 | LReLU | 64 x 32 x 32 |
| Conv 3 x 3 | LReLU | 128 x 32 x 32 |
| Downsample | - | 128 x 16 x 16 |
| Conv 3 x 3 | LReLU | 128 x 16 x 16 |
| Conv 3 x 3 | LReLU | 256 x 16 x 16 |
| Downsample | - | 256 x 8 x 8 |
| Conv 3 x 3 | LReLU | 256 x 8 x 8 |
| Conv 3 x 3 | LReLU | 512 x 8 x 8 |
| Downsample | - | 512 x 4 x 4 |
| Conv 3 x 3 | LReLU | 512 x 4 x 4 |
| Conv 3 x 3 | LReLU | 512 x 4 x 4 |
| Linear | - | 1 |

# D  HYPERPARAMETERS AND OTHER SETTINGS

## D.1  MIXTURE OF GAUSSIAN

We have done 4 experiments for comparison. Our baseline, *Optimistic Adam*, *Consensus Optimization* and *Zero-GP* settings are listed in Table 7, Table 8, Table 9 and Table 10 respectively. Generator has 4 layers with 256 units in each layer and an additional layer that projects into the data space. The discriminator also has 4 layers with 256 units in each layer and a classifier layer on top. ReLU activation function is used after each affine transformation.

Table 7: Settings for Mixture of Gaussians

batch size = 64
discriminator learning rate = 0.0002
generator learning rate = 0.0002
ADAM $\beta_1 = 0.0$
ADAM $\beta_2 = 0.9$
ADAM $\epsilon = 1e - 8$
$\beta = 0.999$ for EMA
max iteration = 40000
GP $\lambda = 1.0$
$n_{dis} = 1$
MA start point = 20000
GAN objective = GAN
Optimizer = ADAM

Table 8: Settings for Mixture of Gaussians for *Optimistic Adam*

batch size = 64
discriminator learning rate = 0.0002
generator learning rate = 0.0002
ADAM $\beta_1 = 0.0$
ADAM $\beta_2 = 0.9$
ADAM $\epsilon = 1e - 8$
$\beta = 0.999$ for EMA
max iteration = 40000
GP $\lambda = 1.0$
$n_{dis} = 1$
MA start point = 20000
GAN objective = GAN
Optimizer = OptimisticADAM

## D.2  CIFAR-10, STL-10, CELEBA, IMAGENET

Table 9: Settings for Mixture of Gaussians for *Consensus Optimization*

batch size = 64
discriminator learning rate = 0.0002
generator learning rate = 0.0002
RMSProp $\beta = 0.9$
RMSProp $\epsilon = 1e - 10$
$\beta = 0.999$ for EMA
max iteration = 40000
Consensus $\gamma = 10.0$
$n_{dis} = 1$
MA start point = 20000
GAN objective = GAN
Optimizer = RMSPropOptimizer

Table 10: Settings for Mixture of Gaussians for *Zero-GP*

batch size = 64
discriminator learning rate = 0.0002
generator learning rate = 0.0002
ADAM $\beta_1 = 0.0$
ADAM $\beta_2 = 0.9$
ADAM $\epsilon = 1e - 8$
$\beta = 0.999$ for EMA
max iteration = 40000
*Zero-GP* $\lambda = 1.0$
$n_{dis} = 1$
MA start point = 20000
GAN objective = GAN
Optimizer = ADAM

Table 11: Settings for CIFAR-10, STL-10, CelebA, ImageNet

batch size = 64
discriminator learning rate = 0.0002
generator learning rate = 0.0002
ADAM $\beta_1 = 0.0$
ADAM $\beta_2 = 0.9$
ADAM $\epsilon = 1e - 8$
$\beta = 0.9999$ for EMA
max iteration = 500000
WGAN-GP $\lambda = 10.0$
WGAN-GP $n_{dis} = 5$
MA start point = 100000
GAN objective = GAN or WGAN-GP
Optimizer = ADAM

