# OpenReview forum: "The Unusual Effectiveness of Averaging in GAN Training"
_ICLR.cc/2019/Conference_

### Official Review · AnonReviewer3 · 2018-10-29
**Interesting sketch of an analysis; mostly an experimental study**

**Rating:** 6
**Confidence:** 4

**Review:**

The submission analyzes parameter averaging in GAN training, positing that using the exponential moving average (EMA) leads to more well-behaved solutions than using moving averages (MA) or no averaging (None).

While reading the submission, the intuitively given explanations for using EMA (cycling, mainly) seem reasonable. However, I do not think there is sufficient understanding of the (non-)convergence behavior in real-world GAN settings, and this submission does not contribute much to it.
The theoretical underpinnings in Section 3.1 are quite thin, and focus on describing one particular example of a bilinear saddle problem, which is quite far from a typical GAN, as used e.g. in computer vision problems. Although interesting to read, I would not draw any wider-reaching conclusions from this carefully constructed example.

Instead, the submission serves mainly as an experimental study on why EMA works better in some of the tested cases than MA/None. Main quantitative measures are the often-used IS and FID. It is clear from both the provided quantitative values as well as the provided qualitative images that either averaging method is likely better then no averaging.

Unfortunately, IS and FID contradict each other somewhat for EMA vs. MA in Table 2, which is attributed to IS being [more] flawed [than FID]. Neither measure is flawless, however, which diminshes the usefulness of the numeric results somewhat. Well designed human studies may be complicated to set up and costly to conduct, but these could demonstrate additional confirmation of the usefulness of the proposed method.

EMA introduces an additional hyperparameter, beta, which is only discussed very briefly, and only in the context of qualitative results. I missed a more thorough discussion of the impact of beta.

Overall, the submission makes an interesting proposition (usage of EMA during GAN training), but falls short in convincing me that this is a useful thing to do in broader contexts. Overall originality is minor; projected significance is minor to medium.

EDIT: After the rebuttal, resulting in several changes and additions to the paper, I am changing my rating from 5 -> 6.

---

> ### Author Response · Authors · 2018-11-26
> **Clarification, new theoretical and empirical results**
>
> Thanks for your review and detailed feedback.
>
> We could not agree more that our field's understanding of the (non-)convergent behavior in real-world GAN settings is lacking. Nevertheless, we should not be evaluated in that direction as our aim is different. We only aim to understand the differential effects of adding to a pre-existing GAN setup, an easy-to-implement averaging technique, argue that this additional step can only help and indeed that EMA in particular consistently and significantly improves system performance.
>
> To better match our stated goal above, and following your suggestions about moving towards typical GAN settings, we have expanded our theoretical analysis with a new section (see section 4.2) that analyzes the local stability of the EMA method for general GAN settings under minimal assumptions on their training algorithms/loss functions. The key theoretical insight is that EMA preserves the stability of locally stable fixed points (Nash equilibria) providing further theoretical evidence in support of averaging. So, indeed as hoped adding EMA only helps the stability of the system.
>
> In our initial submission, we have searched for a good $\beta$ value over range [0.9,0.9999] and used it for all our experiments without stating how good each is. After your suggestion, we have updated the manuscript for CIFAR-10 experiments (in Appendix A.2). We can see a clear and progressive improvement when $\beta$ increases from $0.9$ to $0.9999$ in both IS and FID.
>
> This improvement is in total agreement with the theoretical picture conveyed by our analysis of our small toy example in section 4.1. (see figure 1). So, although this result is clearly meant for illustrative and pedagogical purposes, we nevertheless have found that it to be instructive and even predictive for our hyperparameter search in practice. Since it is a simple and easy to remember example, we hope that it will be similarly helpful for other researchers as well.

---

> > ### Comment · AnonReviewer3 · 2018-12-04
> > **Thank you for the rebuttal**
> >
> > I have adjusted my rating accordingly.

---

### Official Review · AnonReviewer1 · 2018-11-02
**An interesting experimental paper exploring the effect of parameter averaging in GANs**

**Rating:** 6
**Confidence:** 4

**Review:**

This paper tries to adapt the concept of averaging, well known is the game literature, to GAN training. In a simple min-max example the iterates obtained by gradient method do not converge to the equilibrium of the game but their average does. This work first provides intuitions on the potential benefits of exponential moving average (EMA) on a simple illustrative example and explore the effect of averaging on GAN.

In think that the approach of this paper is interesting. I particularly like the experiments on Celeb-A (Fig 6 and 7) that seem to show that the averaged iterates change more smoothly (with respect to the attributes of the faces) during the training procedure. Nevertheless, I have some concerns about the claims of the paper and the experimental process.

I'm surprised by the values of the inception score provided in Table 2 which do not seem to correlate with the sample quality in Fig. 3. Why did not you use the standard implementation of the inception score provided in Salimans et al. [2016]'s paper ?

I think that the effectiveness of EMA over uniform averaging is a bit overclaimed.
- From a theoretical point of view uniform averaging works better (at least in your example in 3.1): If you (uniformly) average the periodic orbit you get a converging iterate. Moreover, concerning to this toy example, note that this continuous analysis has been already introduced in [Goodfellow et al., 2016] and the Hamiltonian interpretation has been already provided in [Balduzzi et al. 2018].
However I think that the intuition on the vanishing magnitude of the oscillation provided by EMA is interesting.
- The continuous dynamics is actually different from the discrete one, I think that an analysis on the discrete case that is used in practice might be more insightful.
- The comparison with uniform averaging is not fair in the sense that uniform averaging has no hyperparameter to tune: In figure 6 uniform averaging performs better than a not well tuned EMA. A fair comparison would be for instance to propose a parametrized online averaging $\theta_{MA}^t = \frac{t - \alpha}{t} \theta_{MA}^{t-1} + \frac{\alpha}{t} \theta_t$ and to tune it the same way $\beta$ is tuned in EMA.

Refs:
Salimans, Tim, et al. "Improved techniques for training gans." Advances in Neural Information Processing Systems. 2016.
Goodfellow, I. (2016). NIPS 2016 tutorial: Generative adversarial networks. arXiv preprint arXiv:1701.00160.
Balduzzi, David, et al. "The Mechanics of n-Player Differentiable Games." ICML (2018).

Minor comments:
- In the introduction "gradient vector fields of the game may not be conservative (Mescheder et al. 2017)" and the related work "Mescheder et al. (2017) states that a reason for non-convergence is the non-conservative gradient
vector of the players.": the notion of conservative vs. non-conservative vector field is never mentioned in [Mescheder et al. 2017]. I think you are actually referring to the blog post on that paper https://www.inference.vc/my-notes-on-the-numerics-of-gans/ .
- In the Related work "can not"
- "In fact, it has recently been established that the smooth (continuous-time) analogues of first order methods such as online gradient descent (follow-the-regularized leader) in bilinear zero-sum games are recurrent (i.e. effectively periodic)
with trajectories cycling back into themselves. " can you provide a citation ?
- Some published papers are refereed as arxiv paper ( for instance (Mescheder et al. 2017) and (Mescheder et al. 2018)), you should cite the published version.

---

> ### Author Response · Authors · 2018-11-26
> **Clarification, new theoretical and empirical results**
>
> Thank you for your review and detailed feedback.
>
> It is great to hear that you have thought that our continuous time analysis of was insightful!
>
> We have added a new discrete time analysis of local stability of EMA techniques when applied to general GAN training algorithms (section 4.2). The theoretical analysis is particularly supportive of the practicality of these techniques, as averaging is shown to preserve the stability of fixed points/Nash equilibria regardless of the details of the training algorithm, structure of the cost functions. This is strongly suggestive of EMA's abilities to work well in a wide range of different datasets/architectures, which we have shown via thorough experimental investigations.
>
> For convenience we have used chainer implementation of Inception score ( https://github.com/mattya/chainer-inception-score/tree/0c43b55b9bcba8149a9ed0b5d0bc4c5eceb49540 ) which has been used in two papers of Miyota Takeru, et. al [2018]. In general samples look cleaner which might be preferred by inception score as the inception network is trained on clean images.
>
> Furthermore, as you have suggested, we have included experiments of parameterized online averaging (see appendix A.1). We see that even though this method tends to increase IS and FID scores w.r.t. non-averaged one, it is still not as good as EMA.
>
> Ref:
> Miyota Takeru, et. all ''Spectral Normalization for Generative Adversarial Networks'', ICLR2018
> Miyota Takeru, et. all ''cGANs with Projection Discriminator'', ICLR2018

---

> > ### Comment · AnonReviewer1 · 2018-12-04
> > **Thank you for these clarifications**
> >
> > Thank you for these clarifications.
> >
> > You local stability analysis is interesting but what it says is that EMA does not change anything in term of second order stability (fortunately it is not worse, but it is also not better).
> > Moreover, your simple bilinear model has pure imaginary eigenvalues and thus the eigenvalues of its Jacobian never lie into the unit ball (whatever the step-size).
> >
> > I'm not sure the discrete local stability analysis you provided is a good point in favor of EMA. (Look like it actually shows that your iterates cannot converge to the equilibrium in the bilinear case)
> >
> > However the experimental approach of this paper remains interesting as I developed it in my review.
> >
> > That is why I will not change my grade.

---

### Official Review · AnonReviewer2 · 2018-11-02
**novelty would be low**

**Rating:** 5
**Confidence:** 2

**Review:**

The paper evaluates two moving average strategies for GAN optimization. Since exact theoretical analysis is difficult for this case, some informal consideration are provided for explanation of performance gain. Experiments confirmed high performance of averaging.

The basic idea seems to be reasonable. Moving average-based strategy would stabilize optimization process.

The obvious weakness of the paper is technical novelty. Although the experimental improvement is confirmed, I would have to say just comparing two known averaging methods would not have strong novelty.

Section 3.1 would be most important part of the paper, but it only mentions quite general tendency of averaging (seems not specific to GAN).

---

> ### Author Response · Authors · 2018-11-26
> **Clarification, new theoretical and empirical results**
>
> Thank you for your review and feedback.
>
> Indeed as you point out theoretical analysis of the setting that we consider is difficult. Nevertheless, we have further expanded our theoretical section (which used to be 3.1) with a new section that performs local stability analysis of EMA techniques for GANs under rather weak assumptions about both the training algorithms and the loss functions (see section 4.2). We believe that by providing both global theoretical analysis of a simple but illustrative example (section 4.1) as well by adding local analysis for a broad range of GAN settings (section 4.2) we give as strong justification as we can for the EMA method from a theoretical perspective.
>
> We are happy that you agree that our thorough testing has indeed confirmed experimentally the improvements offered by this technique.
>
> We respectfully disagree about the lack of technical novelty in our paper. For example, we know of no prior work in the GAN literature that makes theoretical arguments about how system parameters control the magnitude of oscillations. We believe this approach is both conceptually novel and technical interesting.
>
> Moreover, we think that our paper has several counter-intuitive components that are worth noting. Firstly, we prove that averaging in the sense of EMA fails to stabilize even the simplest of toy systems! Secondly, we show that the standard technique of designing training algorithms for GANs around their theoretical performance in bilinear games may be misguided. MA always equilibrates in bilinear games. EMA fails to do so. Hence common practice would suggest to prefer implementing MA over EMA. Our experimental investigation/comparison in a wide range of real GAN settings (e.g. mixture of Gaussians, CelebA, CIFAR-10, STL-10, ImageNet and a host of different algorithms including Adam, Optimistic Adam, Consensus Optimization, etc) shows that the opposite is true. EMA is more robust than MA.

---

### Public Comment · ~Hello_Kitty2 · 2018-10-02
**suggestion**

This work is closely related to the following paper:

Abhay Yadav, Sohil Shah, Zheng Xu, David Jacobs, Tom Goldstein, "Stabilizing Adversarial Nets with Prediction Methods", ICLR 2018.

The authors are encouraged to cite and discuss the differences/contribution of their work.

---

> ### Author Response · Authors · 2018-10-10
> **Reply: suggestion**
>
> Thanks for your comment. The paper tries to alleviate cycling issue in minimax objectives similar to ours, however our approach is totally different then theirs, tough both methods damp oscillations. Their method is more similar to Daskalakis et. al. "Training GANs with Optimism" (https://arxiv.org/abs/1711.00141 ). We will consider to include it.

---

> > ### Public Comment · (anonymous) · 2018-10-10
> > **Link does not exist**
> >
> > The attached link suggested above does not work.

---

### Author Response · Authors · 2018-11-26
**New theoretical and empirical results**

Dear reviewers, thank you for your reviews and feedback. We have greatly expanded our paper with further theoretical and experimental results.

The list of changes includes:
- We have added theoretical analysis of local stability of EMA for general GANs and discrete-time dynamics (see section 4.2).
- We have added an experimental investigation of parameterized variants of online averaging suggested by R1 (see appendix A.1).
- We have added a quantitative study of the effects of parameter \beta in EMA as suggested by R3 (see appendix A.2).
- Due to space constraints, we have moved Figure 2 (the mixture of Gaussians results), Figure 4 (CIFAR-10 FID and IS score), Figure 5 (Comparison of EMA, Optimistic Adam and Consensus Optimization on CIFAR-10), Figure 8 (STL-10 & ImageNet generated datasets) and certain details about the experiments to the Appendix.
- We have addressed other minor comments.

We would be happy to address any remaining questions or concerns!

---

### Meta-Review · Area_Chair1 · 2018-12-14
**Intersting empirical study!**

**Confidence:** 3
**Recommendation:** Accept (Poster)

**Metareview:**

This work analyses the use of parameter averaging in GANs. It can mainly be seen as an empirical study (while also a convergence analysis of EMA for a concrete example provides some minor theoretical result) but experimental results are very convincing and could promote using parameter averaging in the GAN community. Therefore, even if the technical novelty is limited, the insights brought by the paper are intesting.